# Genetic background of high myopia in children

Urh Šenk[1]☯, Bernard Čižman[1]☯, Karin Writzl[1,2], Manca Tekavčič Pompe[1,3]*

1 Faculty of Medicine, University of Ljubljana, Ljubljana, Slovenia, 2 Clinical Institute of Genomic Medicine, University Medical Center Ljubljana, Ljubljana, Slovenia, 3 Eye Hospital, University Medical Centre Ljubljana, Ljubljana, Slovenia

☯ These authors contributed equally to this work.
* manca.tekavcic-pompe@guest.arnes.si

## Abstract

### Objective

High myopia is a significant risk factor for irreversible vision loss and can occur in isolation or as a component of various syndromes. However, the genetic basis of early-onset high myopia remains poorly understood. We aimed to identify the causative genetic variants for high myopia in a cohort of Slovenian children.

### Methods

The study included children referred to a tertiary paediatric ophthalmology centre at the University Eye Clinic in Ljubljana between 2010 and 2022. The participants met the following inclusion criteria: age $\leq$ 15 years and high myopia $\leq$-5.0 D before the age of 10 years. Genetic analysis included exome sequencing and/or molecular karyotyping. Participants were categorized based on clinical presentation: high myopia with systemic involvement, high myopia with ocular involvement, and isolated high myopia.

### Results

Genetic analysis of 36 probands revealed a genetic cause of high myopia in 22 (61.1%) children. Among those with systemic involvement (50.0%), genetic causes were identified in 13 out of 18 children, with Stickler's and Pitt-Hopkins being the most common syndromes. Among cases of high myopia with ocular involvement (38.9%), a genetic cause was found in 8 out of 14 probands, including (likely) pathogenic variants in genes related to retinal dystrophies (*CACNA1F*, *RPGR*, *RP2*, *NDP*). The non-syndromic *ARR3*- associated high myopia was identified in the isolated high myopia group.

### Conclusions

A genetic cause of high myopia was identified in 61.1% of children tested, demonstrating the value of genetic testing in this population for diagnosis and proactive counseling.

**Citation:** Šenk U, Čižman B, Writzl K, Tekavčič Pompe M (2024) Genetic background of high myopia in children. PLoS ONE 19(11): e0313121. https://doi.org/10.1371/journal.pone.0313121

**Data Availability Statement:** All relevant data are within the paper and its Supporting information files.

**Funding:** The author(s) received no specific funding for this work.

**Competing interests:** The authors have declared that no competing interests exist.

## Introduction

Myopia is one of the most common refractive errors, in which images of distant objects are focused in front of the retina, resulting in blurred distance vision [1, 2]. Myopia can be categorised as mild (-0.5 D to -3.00 D), moderate (-3.00 D to -5.00 D or -6.00 D), or high ($\leq$ -5.00 D or -6.00 D) [1]. Some sources also define high myopia (HM) based on an axial length > 26 mm [3, 4]. This classification has clinical significance, as individuals with high myopia face a substantially increased risk of secondary ocular pathologies and irreversible vision loss. Pathologic myopia is a distinct type of HM characterized by degenerative ocular tissue damage associated with myopic maculopathy, retinoschisis, and other complications [1, 2]. Although myopia is often considered a benign condition, it is a major public health problem [2]. Projections show that by 2050, one in two people will be myopic, and one in ten will be highly myopic [5].

While common myopia was once considered primarily genetic, it is now understood as a complex condition arising from the interplay of environmental and genetic factors, with individual genes influencing the severity of the clinical presentation [2, 6]. The familial clustering of myopia, particularly among offspring of myopic parents, highlights the importance of genetic factors in its development. The influence of the genome on the development of the eye, including myopia susceptibility, has also been demonstrated in several twin studies [7]. The stronger familial clustering observed in HM in contrast to low myopia indicates a greater genetic influence on the development of HM [8].

Today, more than 100 different genes and over 20 chromosomal loci are associated with the development of myopia [4]. Some HM cases demonstrate monogenic inheritance, with alterations in a single gene (e.g., *ARR3*, *BSG*, *CTSH*, *CCDC111*, *LEPREL1*, *LOXL3*, *LRPAP1*, *NDUFAF7*, *P4HA2*, *SCO2*, *SLC39A5*, *UNC5D* and *ZNF644*) as the primary cause [1]. Isolated alterations in a single causative gene may also be responsible for many cases of syndromic (high) myopia (i.e., myopia that occurs in association with a specific systemic disability or in combination with at least one other condition). These conditions include a number of intellectual disability syndromes (e.g., Angelman, Bardet-Biedl, Cohen, and Pitt-Hopkins syndromes) and inherited connective tissue disorders such as Marfan, Stickler and some forms of Ehlers-Danlos syndrome [1, 6, 9]. The classification into syndromic and non-syndromic myopia can sometimes be unclear, especially in patients with a syndrome in which the clinical manifestation is limited exclusively to an isolated HM, without additional systemic or ocular features, as occasionally observed in Stickler syndrome, for example [10]. For this reason, a phenotype-based classification of HM into the following groups, used in similar studies, may enhance clinical decision-making: HM with systemic features, HM with ocular involvement, and isolated HM [6, 10].

Recent advances in genetics and genetic testing have enabled the investigation of the genetic background of HM in children in Slovenia, with the potential for improved diagnosis and future therapies. This study aimed to identify the causative genetic alterations (chromosomal rearrangements, point mutations, insertions, deletions, duplications, . . .) that suggest a high causative probability for HM in Slovenian children aged 15 years or younger using exome sequencing (ES) and molecular karyotyping. Additionally, it seeks to determine whether routine genetic testing in the studied population is justified.

## Materials and methods

### The cohort

We used data from children referred to the tertiary pediatric ophthalmology center at University Eye Clinic in Ljubljana between 2010 and 2022 who met the following inclusion criteria:

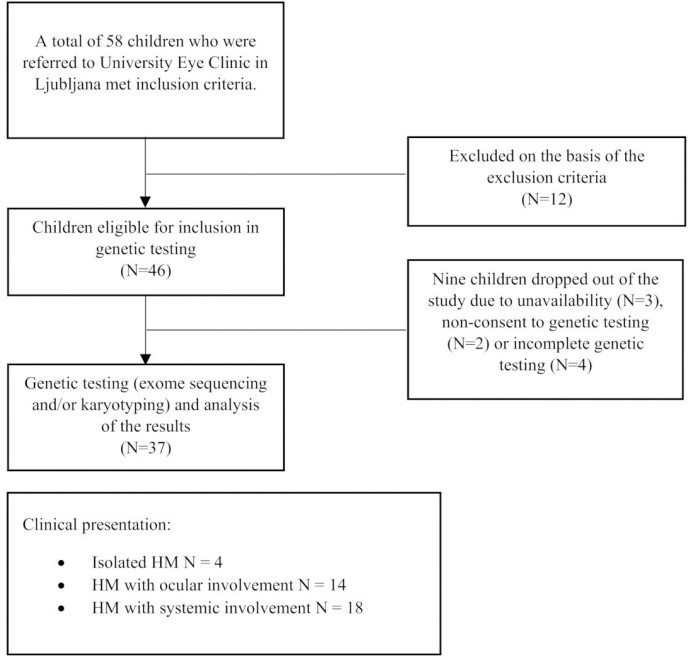

**Fig 1. The flowchart outlining participant enrollment, exclusion, and attrition.**

age up to and including 15 years and HM equal to or exceeding SE -5.0 D before the age of 10 years. We excluded children who were highly likely to have developed HM as a result of another eye condition, e.g., retinopathy of prematurity, congenital glaucoma, secondary glaucoma after early surgery for congenital cataract, other types of secondary glaucoma, and exacerbation or development of HM as a complication of surgical treatment of other ocular pathology. Children with HM who underwent genetic analysis as part of segregation analysis but were not the probands were excluded. A flowchart of children who participated in the study is presented in Fig 1.

Written informed consent was obtained from the parents of all participating children. The research adhered to the tenets of the Declaration of Helsinki and was approved by the Republic of Slovenia National Medical Ethics Committee (case number 0120-284/2022/3).

## Genetic testing

The results of the genetic tests were reviewed at the Clinical Institute of Genomic Medicine (CIGM) and the Genetic Clinic of the Children's Clinic in Ljubljana. Patients with non-syndromic high myopia (HM) underwent exome sequencing (ES). In syndromic HM cases, molecular karyotyping was performed first, with ES used subsequently if no causative mutation was identified. All genetic testing was performed as part of routine diagnostics. The DNA was isolated from peripheral blood samples using the Qiagen Mini kit (Qiagen, Valencia, CA). Microarray analysis was performed using the oligonucleotide array Agilent Technologies 4 × 180 K, according to the manufacturer's instructions. The array images were acquired using the Agilent laser scanner G2565CA. The image files were quantified using the Agilent Feature extraction software and analyzed with the Agilent Cytogenomics 5.3 software (Agilent Technologies). Interpretation of the microarray results was performed according to the ACMG and ClinGen guidelines [11], taking into account publicly available databases (DECIPHER

(Database of Chromosomal Imbalance and Phenotype in Humans using Ensembl Resources https://decipher.sanger.ac.uk/), Database of Genomic Variants (DGV) (http://dgv.tcag.ca/dgv/app/home), ClinVar (https://www.ncbi.nlm.nih.gov/clinvar/), ClinGen (http://dbsearch.clinicalgenome.org/search/)), and the in-house database of detected variants [12]. Exome sequencing was performed using a standardized series of procedures, starting with an in-solution capture of exome sequences (TruSight One, TruSight Exome, and Nextera Coding Exome capture kits, Agilent SureSelect Human All Exon v2, Agilent SureSelect Human All Exon v5 capture kits). This was followed by sequencing on Illumina MiSeq or Illumina HiSeq 2500 platform. Exome data analysis was conducted using an internally developed pipeline along with various bioinformatics approaches, including phenotype-driven gene targets and trio design, as previously described [13, 14]. Briefly, each variant was annotated based on data such as chromosomal location, zygosity, transcript information, predicted effects on protein structure and function, and local and global frequency data from The Genome Aggregation Database (gnomAD) and the Slovenian Genomic Database. In silico predictions and previous associations with diseases were sourced from the HGMD, LOVD, and ClinVar databases, as well as through literature searches. Variants were categorized as pathogenic, likely pathogenic, or variant of uncertain significance according to the American College of Medical Genetics and Genomics (ACMG) and Association for Molecular Pathology standards for interpreting sequence variants [15]. Variants classified as pathogenic or likely pathogenic were labeled causative if they were associated with patient phenotypes.

In patients with isolated HM and those with HM and other ocular involvement, an eye disorder gene panel comprising 549 genes was used (S1 Table). For patients with HM and systemic involvement, this panel was supplemented with a phenotype-based gene panel generated using a web tool (https://www.kigm.eu/generator/). This tool facilitated tracking genes associated with clinical signs and symptoms using HPO nomenclature [13, 14]. If results were negative, an untargeted interpretation of all exome variants was performed.

## Phenotyping

The children were examined by a pediatric ophthalmologist and a clinical geneticist and categorized into three groups based on their clinical presentation: HM with systemic involvement, HM with other ocular involvement, and isolated HM. Children with HM and systemic features with or without additional ocular pathologies were classified in HM with systemic involvement group, while the HM with other ocular involvement group included children in which HM was accompanied with additional pathologies limited to the eye (e.g., retinal dystrophies, strabysmus, anisometric amblyopia, nystagmus, . . .).

## Statistics

Thirty-seven children who underwent genetic testing were included in the analysis. Demographic data (age and sex) and refractive error are presented as average value ± standard deviation (SD). Basic calculation software was used to perform statistical analysis. Statistical calculations were based on 36 probands, as two children were siblings sharing the same genetic variant. The cohort was divided into four groups based on the result of the genetic testing: those with confirmed genetic causes for HM, those with variants of uncertain significance (VUS), those with (likely) pathogenic genetic variants related to a syndromic phenotype but not previously linked to HM development, and those lacking identifiable genetic variants to explain their clinical phenotype. Percentages of genetically confirmed probands were calculated and presented, divided in systemic, ocular involvement and isolated high myopia group.

## Results

A total of 46 children (24 boys and 22 girls) were included in the study. The mean age of the children at inclusion in the study was 8.9 ± 3.7 years, and the mean refractive error was -9.1 ± 3.1 D. Nine children were lost to follow-up due to unavailability, non-consent to genetic testing, or incomplete genetic analysis.

Of the 36 probands, we confirmed a genetic cause of HM in 22 children (61.1%). Two children (5.6%) were found to have genetic variants classified as VUS, and four children (11.1%) had pathogenic or likely pathogenic variants in genes not previously associated with the development of myopia. In eight children (22.2%), we could not identify causative genetic variants for the distinct clinical phenotype. Table 1 shows the numerical overview of genetic testing for children in each group. Detailed information regarding clinical presentation and genetic testing for individual children can be found in Table 2.

We identified a genetic cause in 13 of 18 probands with a clinical picture of HM in the context of systemic involvement. A monogenic cause was identified in eight of these children and a chromosomal cause in five. Stickler syndrome (N = 2; causative variant in the *COL2A1* gene) and Pitt-Hopkins syndrome (N = 3; one causative variant in the *TCF4* gene, two cases associated with 18q deletion syndrome) were most common in this cohort. In our sample, we also associated one case each of Noonan syndrome (causative variant in the *BRAF* gene), Knobloch syndrome (causative variants in the *COL18A1* gene), Marfan syndrome (causative variant in the *FBN1* gene), MEB disease (causative variants in the *POMGNT1* gene), CHARGE syndrome (causative variant in the *CHD7* gene), Down syndrome (trisomy 21), Prader-Willi syndrome (microdeletion of the 15q11.2 region), and 8q13 microdeletion syndrome.

In one (1/18) child, we found genetic variants defined as VUS, and in four (4/18) children, we found (likely) pathogenic variants in the genes *TRPV4*, *TUBA1A*, *SBDS*, *KIDINS220*, which have not been previously associated with the development of myopia.

Fourteen probands (38.9%) had a clinical presentation of HM with ocular involvement, with a monogenic cause identified in 8 of the 14 cases. Specifically, HM was identified in six children with retinal dystrophies (*RPGR*; N = 2, *CACNA1F*; N = 2, *RP2*, *NDP*) and one child each of corneal dystrophy (*ZEB1*) and Stickler syndrome (causative variant in the *COL9A2* gene).

In the group of probands with isolated HM (N = 4; 11.1%), we identified a genetic cause in one out of four probands. Specifically, a pathogenic variant in *ARR3* was found in two siblings.

## Discussion

The genetic analysis of the 36 probands revealed a genetic background for the clinical picture of HM in 61.1% of the children. HM manifested with systemic features in 50.0% of probands, with ocular involvement in 38.9%, and as an isolated disease in 11.1%.

**Table 1. Numerical overview of genetic test results for all probands and their further subdivision according to subgroup.**

|  | Number of probands (%) | Confirmed genetic cause of HM (%) | VUS (%) | Other genetic variant not associated with the development of HM (%) | No genetic variant to explain the clinical phenotype (%) |
|---|---|---|---|---|---|
| **Total sample** | 36 (100%) | 22 (61.1%) | 2 (5.6%) | 4 (11.1%) | 8 (22.2%) |
| **HM with systemic involvement** | 18 (50.0%) | 13 (36.1%) | 1 (2.8%) | 4 (11.1%) | 0 (0.0%) |
| **HM with ocular involvement** | 14 (38.9%) | 8 (22.2%) | 1 (2.8%) | 0 (0.0%) | 5 (13.9%) |
| **Isolated HM** | 4 (11.1%) | 1 (2.8%) | 0 (0.0%) | 0 (0.0%) | 3 (8.3%) |

VUS—variant of uncertain significance, HM—high myopia

**Table 2. Detailed ophthalmologic characteristics and genetic testing results in our cohort.**

| | Age (years) | Refractive error (D) | Signs of pathological myopia | Genetic variant | Type of zygosity | ACMG classification | Diagnosis |
|---|---|---|---|---|---|---|---|
| 1 | 5,7 | R: -1,00/30˚ L: -8,50-1,50/140˚ | Peripapillary atrophy, signs of retinal thinning and eyeball elongation, L macula without reflection | arr[GRCh37] 9p24.3p24.2 (2076268_2737525)x3 mat, 16p13.2 (8768819_9598226)×3∼4 | Heterozygote | VUS | / |
| 2 | 8,6 | RL: -8,50-1,50/180˚ | Papillae with hypoplastic ring, thin retinal vasculature, pigment dispersion, dystrophic fundus | NM_004333.4(BRAF): c.1502A>G p.(Glu501Gly) | Heterozygote | Likely pathogenic | Noonan syndrome |
| 3 | 5,4 | RL: -8,00 | Peripapillary atrophy, retinal thinning | NM_021625.5(TRPV4): c.805C>T p.(Arg269Cys) | Heterozygote | Pathogenic[a] | Congenital distal spinal muscular atrophy |
| 4 | 4,8 | R: -7,00 | Rarefied pigment | trisomy chromosome 21 | / | Pathogenic | Down's syndrome |
| | | L: -6,00 | | | | | |
| 5 | 4,1 | RL: -13,00 | R: Retinal thinning; | NM_130444.3(COL18A1): c.2174-2A>G | Compound heterozygote | Likely pathogenic | Knobloch syndrome |
| | | | L: Fundus fully hypopigmented | NM_130444.3(COL18A1): c.4759_4760delCT p.Leu1587fs | | Pathogenic | |
| 6 | 9,4 | R: -7,00-0,75/180˚ | Peripapillary atrophy | NM_001270399.1(TUBA1A): c.641G>A p.(Arg214His) | Heterozygote | Likely pathogenic[a] | TUBA1A tubulinopathy |
| | | L: -7,00-1,50/180˚ | | | | | |
| 7 | 9,0 | R: -11,25-1,5/46˚ | Thinned chorioretina Temporal zone without vascularisation | NM_016038.2(SBDS): c.183_184delinsCT (p.Lys62*), c.258+2T>C | Compound heterozygote | Pathogenic[a] | Shwachman-Diamond syndrome |
| | | L: -10,25-1,5/154˚ | | | | | |
| 8 | 9,2 | RL: -14,00 | Peripapillary atrophy, retinal pigmentation | arr[GRCh37] 18q21(47,782,339-60,231,879)x1 | Heterozygote | Pathogenic | Pitt-Hopkins syndrome |
| 9 | 10,8 | R: -10,00 | / | NM_000138.4(FBN1): c.1481G>T p.(Cys494Phe) | Heterozygote | Likely pathogenic | Marfan syndrome |
| | | L: -7,00 | | | | | |
| 10 | 11,1 | RL: -6,00 | / | NM_017739.4(POMGnT1): c.1285-2A>G, c.1539+1G>A | Compound heterozygote | Pathogenic | Muscle-eye-brain disease |
| 11 | 11,3 | RL: -6,00 | Peripapillary atrophy, retinal thinning | arr[GRCh19] 8q13.2.q13.3(68,109,818-71,102,561)x1 | Heterozygote | Pathogenic | 8q13 microdeletion syndrome |
| 12 | 7,2 | RL: -9,00 | Peripapillary atrophy, posterior staphyloma | arr[hg19] 18q21.1.q22.3(52,932,969-68,873,702)x1 | Heterozygote | Pathogenic | 18q deletion syndrome (includes *TCF4*) |
| 13 | 6,3 | RL: -10,00 | / | NM_017780.3(CHD7): c.4353+1G>A | Heterozygote | Pathogenic | CHARGE syndrome |
| 14 | 13,2 | R: -10,75-6/25˚ | Retinal thinning | NM_001083962.1(TCF4): c.1832T>G p.(Leu611Arg) | Heterozygote | Likely pathogenic | Pitt-Hopkins syndrome |
| | | L: -12,75-4,75/160˚ | | | | | |
| 15 | 5,5 | R: -6,50-2,50/180˚ | / | NM_001348729.2(KIDINS220): c.4056+1G>T | Heterozygote | Likely pathogenic[a] | Spastic paraplegia-intellectual disability-nystagmus-obesity syndrome |
| | | L: -7,00-2,00/180˚ | | | | | |
| 16 | 12,8 | R: -13,50 | Bilateral retinal detachment (R in 2020, L in 2017)) | NM_149162.2(COL2A1): c.2503C>T p.(Arg835Cys) | Heterozygote | Pathogenic | Stickler syndrome |
| | | L: -10,00 | | | | | |
| 17 | 10,0 | R: -6,00-1,50/30˚ | / | NM_001844.5(COL2A1): c.2111dupC p.(Gly705fs) | Heterozygote | Likely pathogenic | Stickler syndrome |
| | | L: -5,50-1,50/120˚ | | | | | |
| **18** | 13,2 | R: -5,50-1,50/180˚ | / | rsa 15q11.2(Me028)x1 Methylation pattern: paternal allele absent | Heterozygote | Pathogenic | Prader-Willi syndrome |
| | | L: -4,50-1,50/180˚ | | | | | |

*(Continued)*

**Table 2.** (Continued)

| | Age (years) | Refractive error (D) | Signs of pathological myopia | Genetic variant | Type of zygosity | ACMG classification | Diagnosis |
|---|---|---|---|---|---|---|---|
| 19 | 11,2 | R: -15,75-1,00/ 90° <br> L: -8,50-0,75/ 90° | Peripapillary atrophy | NM_001852.4(COL9A2): c.803G>A p.(Gly268Asp) | Heterozygote | VUS | / |
| 20 | 0,4 | RL: -7 | Thinned retina | NM_001844.5(COL2A1): c.3706delC p.(Leu1236CysfsTer13) | Heterozygote | Pathogenic | Stickler syndrome |
| 21 | 9,1 | R: -6,50-3,50/ 4° <br> L: -8.00-2,50/ 178° | / | NM_001174096.1(ZEB1):c.238A>T p.(Lys80*) | Heterozygote | Likely pathogenic | Corneal dystrophy |
| 22 | 5,2 | R: -13,00-2,0/ 160° <br> L: -12,50-2,0/ 160° | Peripapillary atrophy, retinal thinning | NM_001034853.2(RPGR): c.2236_2237delGA p.(Glu746fs) | Hemizygote | Pathogenic | Retinal dystrophy |
| 23 | 14,8 | R:-5,75-3,50/ 10° <br> L: -5,00-2,50/ 176° | Bilateral peripapillary atrophy | Exome sequencing did not reveal the presence of genetic variants that could explain the clinical picture. | / | / | / |
| 24 | 8,5 | R: -14,75-2,25/ 10° <br> L: -14,25-1,25/ 150° | Peripapillary atrophy, thinned and dystrophic chorioretinas | NM_001256789.3c(CACNA1F): c.5446C>T p.(Arg1816Ter) | Hemizygote | Pathogenic | Dystrophy of the rods and cones |
| 25 | 7,8 | R: -10,00 <br> L: -12,00 | Retinal thinning | NM_000266.3(NDP):c.414C>T p.(Ala105Val) | Hemizygote | Pathogenic | Norrie's disease |
| 26 | 8,9 | R: -10,00-1,50/ 10° <br> L: -9,50-1,50/ 170° | / | NM_001034853.2(RPGR):c.2977G>T p.(Glu993*) | Heterozygote | Likely pathogenic | Retinal dystrophy |
| 27 | 12,1 | R: -7,00-2,50/ 10° <br> L: -6,50-3,0/5° | Retinal thinning | NM_006915.2(RP2): c.358C>T p.(Arg120Ter) | Hemizygote | Pathogenic | Retinal dystrophy |
| 28 | 14,8 | R: -17,25-1,75/ 30° <br> L:-16,00-2,25/ 80° | Postoperative retinal detachment L, peripapillary atrophy, retinal thinning | Exome sequencing did not reveal the presence of genetic variants that could explain the clinical picture. | / | / | / |
| 29 | 13,4 | R: -6,00 <br> L: -8,00 | / | Exome sequencing did not reveal the presence of genetic variants that could explain the clinical picture. | / | / | / |
| 30 | 11,3 | R: -9,00-2,75/ 5° <br> L: -9,25-3,25/ 160° | Peripapillary atrophy | NM_005183.2(CACNA1F):c.448G>C p.(Gly150Arg) | Hemizygote | Pathogenic | Incomplete congenital stationary night blindness |
| 31 | 3,9 | R: -7,00-1,00/ 80° <br> L: -7,00-1,00/ 18° | / | Exome sequencing did not reveal the presence of genetic variants that could explain the clinical picture. | / | / | / |
| 32 | 11,3 | R: -9,0 <br> L: -7,0-2,0/90° | Retina of rarefied pigmented appearance, papillae pale, hypoplastic | Exome sequencing did not reveal the presence of genetic variants that could explain the clinical picture. | / | / | / |
| 33[b] | 4,8 | R: -9,00-2,0/1° <br> L: -10,00-0,75/ 28° | / | NM_004312.3(ARR3):c.214C>T p.(Arg72*) | Heterozygote | Pathogenic | X-linked myopia type 26 |

*(Continued)*

**Table 2.** (Continued)

| | Age (years) | Refractive error (D) | Signs of pathological myopia | Genetic variant | Type of zygosity | ACMG classification | Diagnosis |
|---|---|---|---|---|---|---|---|
| 34[b] | 7,3 | R: -8,0-2,0/5˚, L: -8,0-2,0/180˚ | / | NM_004312.3(ARR3):c.214C>T p. (Arg72*) | Heterozygote | Pathogenic | X-linked myopia type 26 |
| 35 | 3,6 | R: -7,50-2,00/120˚ L: -7,00-1,00/120˚ | Retinal thinning | Exome sequencing did not reveal the presence of genetic variants that could explain the clinical picture. | / | / | / |
| 36 | 4,9 | R: -14,00 L: -15,00 | Retinal thinning | Exome sequencing did not reveal the presence of genetic variants that could explain the clinical picture. | / | / | / |
| 37 | 14,3 | R: -7,00-1,00/40˚ L: -4,00 | / | Exome sequencing did not reveal the presence of genetic variants that could explain the clinical picture. | / | / | / |

Children are classified into groups based on clinical presentation. White coloured cells represent HM with systemic features, blue colour corresponds to HM with ocular involvement and grey colour to isolated HM. D—dioptre, R—right eye, L—left eye, RL—right and left eye

[a] variant not associated with the development of HM

[b] from same family

## Diagnostic yield of the study

The diagnostic yield of our study was higher than expected at 61.1%. Similar studies in this field report a proportion of genetically confirmed HM cases of around 20% and up to 80% in the case of specific subject selection [10, 16–19]. The high diagnostic yield of our study is most likely due to the sample selection. Namely, we included children with HM who were treated in a tertiary paediatric ophthalmology centre, so a higher proportion of pathological conditions can be expected than in the general population of children of the same age. As a result, it is difficult to extrapolate the results of the survey to the general population of children in Slovenia. This suggests that the proportion of genetically explainable cases of HM in children is probably lower in the general population. However, given the high diagnostic yield, it can be argued that referral of children with HM treated in tertiary-level ophthalmology outpatient clinics for genetic testing is reasonable, as this is likely to clarify the aetiology of their clinical condition.

## HM with systemic involvement and HM with other ocular involvement

The percentage of participants in groups with systemic involvement or with ocular involvement varies widely between studies. While Marr et al. (2001) reported 92% of HM cases associated with systemic or ocular features, the hospital-based sampling limits the study's generalizability [20]. In contrast, other investigators have reported a significantly lower percentage of HM cases with systemic or ocular involvement, ranging from 27% to 44% [10, 21]. In particular, Marfan syndrome and Stickler syndrome, in which HM is a common clinical manifestation, occurred frequently in this group. Notably, the potential for Stickler syndrome to initially present as isolated HM highlights the difficulty of phenotypic classification, as other ocular features may manifest later [22].

In our study, 50% of children had HM with systemic involvement, which is significantly higher than the 15% reported by Haarman et al. [10]. The high proportion of systemic disabilities in our sample reflects the elevated prevalence of these conditions among children treated

in an at-risk outpatient clinic. On the other hand, our result is very similar to that of Marr J.E. et al., who found 54% of participants to have an underlying systemic disease with or without other eye problems [20]. In the same study, a similar percentage was found to have HM with other eye problems (e.g., lens subluxation, coloboma, retinal dystrophy, anisometropic amblyopia), namely 38% compared to 38.9% in our study.

Among the syndromes in which HM is common, Stickler's and Knobloch's syndromes are the most frequently described in the literature, which is also consistent with the results of our study [10, 17, 18]. In our sample, we also found HM associated with several other syndromes, and it is particularly interesting that, in contrast to the aforementioned studies, we reported three children (13.6% of genetically confirmed HM cases) with Pitt-Hopkins syndrome.

It is also worth noting that our study describes cases of HM in individuals with variants in genes that have not yet been clearly associated with the development of myopia (*TRPV4*, *TUBA1A*, *SBDS*, *KIDINS220*).

The *TRPV4* gene encodes for a transient receptor potential (TRP) channel that is expressed in all ocular tissues and is critical for maintaining proper ocular physiology. Disturbances in its expression are mainly associated with skeletal dysplasia and neuromuscular disorders, such as congenital distal spinal muscular atrophy, which was the main phenotype of the proband 3, in addition to HM. While myopia has not been previously reported in individuals with TRPV4-related disorders, its potential role in the development of eye diseases such as glaucoma and retinopathy has already been described [23].

*TUBA1A* encodes for α-tubulin, a major component of microtubules, which are involved in essential cellular processes of intracellular transport, cell division and neuronal migration. Pathogenic variants in *TUBA1A* have been shown to cause a neurodevelopmental syndrome, characterized by structural brain malformations, congenital microcephaly, developmental delay, intellectual disability, and epilepsy (OMIM: 611603) [24]. Proband 6 carries a *de novo* heterzygous pathogenic variant c.641G>A, p.(Arg214His) in *TUBA1A*. Her phenotype includes hypotonia, severe global developmental delay, intellectual disability, epileptic encephalopathy, acquired microcephaly, and HM. Tubulin-related genes are essential for brain and eye development. TUBA1A is expressed in the developing eye and has been shown to be essential for the optic nerve regeneration in the zebrafish model [25]. Patients with pathogenic *TUBA1A* variants and various eye anomalies such as coloboma (PMID: 26130693, PMID: 38502138), microphthalmia and cataracts (PMID: 26294046) have been reported, demonstrating the importance of this gene in eye development, structure and function [26–28].

The *KIDINS220* gene encodes a scaffolding transmembrane protein that controls neuronal cell survival, differentiation into exons and dendrites, and synaptic plasticity. *De novo* loss-of-function variants involving the last 2 exons (29 and 30) of *KIDINS220* have been associated with spastic paraplegia, intellectual disability, nystagmus, and obesity (SINO) (OMIM #617296) [29]. The phenotype of the proband 15 in our study with the *de novo likely pathogenic* variant (c.4053+1G>T) corresponded to SINO syndrome, and the ocular symptoms included nystagmus, strabismus, amblyopia, high myopia, and astigmatism. Interestingly, a similar ophthalmologic phenotype was also observed in her identical twin sister with SINO syndrome, who did not participate in the study because her myopia was below the threshold for inclusion in the study. Expression studies during zebrafish and Xenopus laevis embryogenesis have demonstrated dynamic expression of KIDINS220 in the eye region. Additionally, ophthalmologic phenotypes, including nystagmus, reduced vision, and often esotropia, possibly secondary to hypermetropia, have already been described in individuals with SINO syndrome, indicating a role for the gene in eye function [29]. Although pathogenic variants in the *KIDINS220* have not yet been linked to the development of myopia and therefore cannot be considered

causative for this phenotype, the number of individuals SINO syndrome is still small, suggesting that the complete spectrum of ophthalmologic phenotypes may not yet be fully understood.

In the group with only ocular involvement, we observed 6 cases of (probably) pathogenic variants in genes associated with retinal dystrophies (*RPGR*; N = 2, *CACNA1F*; N = 2, *RP2*, *NDP*), which corresponds to 27.3% of all genetically confirmed cases of HM. This is a lower but comparable proportion to the study by Haarman et al. in which pathogenic variants in genes associated with retinal dystrophies were found in 39% of genetically confirmed HM cases (*FAM161A*, *GUCY2D*, *PDE6H*, *CACNA1F*, *NYX*, *RPGR* and *TRPM1*) [10]. A novel pathogenic *NDP* variant was identified in a patient with Norrie's disease. However, a variant affecting the same amino acid site but replacing it with a different amino acid has been previously reported in a patient with Norrie's disease [30]. The *RP2* variant found is the most frequently reported pathogenic variant in the *RP2* gene in patients with X-linked retinitis pigmentosa [31, 32]. Causal alterations in the *CACNA1F* gene are responsible for incomplete congenital stationary night blindness 2A (OMIM #300071) and retinal dystrophy (OMIM #300476). Both variants found in our study have already been described in the literature [33, 34]. Pathogenic variants in the *RPGR* gene are a known cause of X-linked retinal dystrophy in male patients, which also includes X-linked cone and rod dystrophy 1 (OMIM #304020) and retinitis pigmentosa 3 (OMIM #300029).

In addition, we identified a pathogenic heterozygous variant in the *ZEB1* gene that is a known cause of posterior polymorphous corneal dystrophy 3 (OMIM #609141) and Fuchs endothelial corneal dystrophy 6 (OMIM #613270).

### Isolated form of HM

Our finding of a low prevalence of isolated high myopia (HM) aligns with the 8% reported by Marr et al. [20]. They concluded that the diagnosis of isolated HM can only be made by a process of elimination and that in the case of apparently isolated HM in the first 10 years of life, possible systemic disease should be considered rather than isolated HM, given the significantly higher incidence of ocular and systemic abnormalities associated with HM [20].

We identified a pathogenic variant in the *ARR3* gene occurring in female siblings, a variant previously reported in female myopia patients in two large families [35–37]. Mutations in *ARR3* are recognized as a common cause of isolated HM with X-linked inheritance and female-restricted HM [10, 18].

### Strengths, limitations, and relevance of the study

The study's limited sample size, selective participant recruitment, and short duration necessitate cautious interpretation of findings, particularly regarding broader epidemiological implications. A large-scale, national study is needed to obtain a more accurate picture of HM genetic background in the Slovenian pediatric population. This study should include a broader spectrum of children with HM and reduce potential sampling biases seen in our tertiary-care setting.

### Conclusions

Our study employed molecular karyotyping and exome sequencing to investigate the genetic basis of HM in children aged 15 years or younger. We identified genetic causes for HM in a significant proportion of children. The diagnostic yield (61.1%) exceeded initial expectations but likely reflects the study's focus on patients from a tertiary care setting. Nevertheless, the high diagnostic performance suggests that genetic testing in children with myopia treated in

specialised clinics may be important to identify the underlying genetic aetiology, as HM may be the first indication of a more complex disorder (e.g., Stickler's or Marfan syndrome). Overall, our findings advance the understanding of the genetic background of HM, highlighting the value of genetic testing for diagnosis, counseling, and potential preventive interventions.

## Supporting information

**S1 Table. Eye disorder gene panel.**
(DOCX)

## Acknowledgments

The authors thank Chiedozie Kenneth Ugwoke, MD, for assistance in proofreading of our manuscript and additional scientific insights. We also thank List d.o.o. (Computer Engineering) for technical assistance with data extraction from the hospital information system.

## Author Contributions

**Conceptualization:** Karin Writzl, Manca Tekavčič Pompe.

**Data curation:** Karin Writzl, Manca Tekavčič Pompe.

**Formal analysis:** Urh Šenk, Bernard Čižman.

**Investigation:** Urh Šenk, Bernard Čižman.

**Methodology:** Karin Writzl, Manca Tekavčič Pompe.

**Supervision:** Karin Writzl, Manca Tekavčič Pompe.

**Validation:** Karin Writzl, Manca Tekavčič Pompe.

**Writing – original draft:** Urh Šenk, Bernard Čižman.

**Writing – review & editing:** Karin Writzl, Manca Tekavčič Pompe.

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
