## [Decision Letter · Decision Letter 0]

28 Jun 2024

PONE-D-24-12970Genetic background of high myopia in childrenPLOS ONE

Dear Dr. Tekavčič Pompe,

Thank you for submitting your manuscript to PLOS ONE. After careful consideration, we feel that it has merit but does not fully meet PLOS ONE’s publication criteria as it currently stands. Therefore, we invite you to submit a revised version of the manuscript that addresses the points raised during the review process.

 Please edit the manuscript as suggested by the reviewers in a precise and accurate manner which is then reported in the response letter point by point.Please ensure that your decision is justified on PLOS ONE’s publication criteria and not, for example, on novelty or perceived impact.

We look forward to receiving your revised manuscript.

Kind regards,

Giuseppe Novelli

Academic Editor

PLOS ONE

Additional Editor Comments (if provided):

Reviewers' comments:

Reviewer's Responses to Questions

**Comments to the Author**

1. Is the manuscript technically sound, and do the data support the conclusions?

Reviewer #1: Partly

Reviewer #2: Partly

2. Has the statistical analysis been performed appropriately and rigorously? 

Reviewer #1: No

Reviewer #2: No

3. Have the authors made all data underlying the findings in their manuscript fully available?

Reviewer #1: No

Reviewer #2: No

4. Is the manuscript presented in an intelligible fashion and written in standard English?

Reviewer #1: Yes

Reviewer #2: No

5. Review Comments to the Author

Reviewer #1: The manuscript was presented as a "research article" although it lacked some objectives such as a description of the procedures, the information used to obtain the results and the implications of the results produced. The introduction gives a fair overview of high myopia. The objective of the manuscript is not clear, therefore it is recommended to describe it in more detail.

In the "Genetic Testing" section, it is recommended to include details of the biological source used for the DNA extraction and molecular experiments as well as notes on ES and molecular karyotyping protocols. Were virtual panels used in the analysis of ES data? If yes, a list of genes should be provided. It is also important to clarify the workflow used for data analysis. In the "Statistics" section, the authors must describe the methods used and the use of any calculation software if it has been employed.

The authors need to clarify whether the syndromic status of the patients was highlighted beforehand. This is a crucial point in defining the objectives of the paper. In the discussion, the part describing "new" genes must be supported by specific clinical evidence that can exclude/include the presence of disease phenotypes. These data are crucial for the clarification of the relationship between high myopia and the different variants identified.

Reviewer #2: The authors present an interesting genetic study in myopic young patients. Some comments are reported below:

1. This part should be removed from the statistic paragraph:

"A total of 46 children (24 boys and 22 girls) were included in the study. The mean age of the children

119 at inclusion in the study was 8.9 ± 3.7 years, and the mean refractive error was -9.1 ± 3.1 D. Nine

120 children were lost to follow-up due to unavailability, non-consent to genetic testing, or incomplete

121 genetic analysis."

And included in the results paragraph.

There is no statistical analysis reported in this study and no statistical tests have been reported but simply the results of the study have been reported.

Lisa

6. PLOS authors have the option to publish the peer review history of their article (what does this mean?). If published, this will include your full peer review and any attached files.

Reviewer #1: No

Reviewer #2: No

---

## [Author Response · Author response to Decision Letter 0]

19 Sep 2024

We are writing to resubmit a revised manuscript titled “Genetic background of high myopia in children” for publication in PLOS ONE. Reviewers' valuable comments and suggestions are greatly appreciated and have undoubtedly improved the quality of our work.

In response to the reviewers' feedback, we have prepared a detailed table addressing each comment and outlining the specific changes made in the manuscript. Additionally, we have made specific revisions directly in the text of the manuscript, which we have also attached.

We are confident that these revisions have strengthened our manuscript and we look forward to your feedback.

Reviewers' comments and author's response (R)

Reviewer 1: “ The manuscript was presented as a "research article" although it lacked some objectives such as a description of the procedures, the information used to obtain the results and the implications of the results produced. The introduction gives a fair overview of high myopia.” 

R: As suggested the description of the procedures was provided in more comprehensive way, we additionally specified the sources and nature of the information used to obtain our results and expanded our discussion on the implications of the results. The "Genetic Testing" section has been rewritten in a more detailed and comprehensive manner.

Reviewer 1: “The objective of the manuscript is not clear, therefore it is recommended to describe it in more detail.” 

R: As suggested the objective of the manuscript was made more comprehensive: “This study aimed to identify the genetic alterations (chromosomal rearrangements, point mutations, insertions and deletions, duplications, …) that suggest a high causative probability for HM in Slovenian children aged 15 years or younger using exome sequencing (ES) and molecular karyotyping.”

Reviewer 1: “In the "Genetic Testing" section, it is recommended to include details of the biological source used for the DNA extraction and molecular experiments as well as notes on ES and molecular karyotyping protocols. Were virtual panels used in the analysis of ES data? If yes, a list of genes should be provided.”

R: We have revised our work to include a more detailed description of the genetic protocols and methods used. The "Genetic Testing" section has been rewritten in a more detailed and comprehensive manner.

Reviewer 1: “It is also important to clarify the workflow used for data analysis.” 

R: Workflow used for data analysis was presented more clearly with a flow-chart, that was already added as a separate file.

Reviewer 1: “In the "Statistics" section, the authors must describe the methods used and the use of any calculation software if it has been employed.” 

R: We have updated the "Statistics" section to include a more detailed description, in order to reflect our approach and provide more clarity: “Demographic data (age and sex) as well as refractive error are shown as average value ± standard deviation (SD). Basic calculation software was used for numerical/statistical analysis.Percentages of genetically confirmed probands were calculated and presented, divided in systemic, ocular involvement and isolated high myopia group.”

Due to the nature and design of our research, performing statistical tests beyond the description of demographic data was not deemed appropriate or meaningful.

Reviewer 1: “The authors need to clarify whether the syndromic status of the patients was highlighted beforehand.” 

R: The syndromic status of the patients was determined based on data available in the databases at the time of data capture, the most recent ophthalmological examination results and the latest genetic testing results. As noted, the classification into syndromic and non-syndromic myopia can sometimes be ambiguous, particularly in cases where a syndrome manifests solely as isolated HM without additional systemic or ocular features, as occasionally observed for example in Stickler syndrome. For this reason, a phenotype-based classification of HM into the following groups, used in similar studies was also utilized in our study: HM with systemic features, HM with ocular involvement, and isolated HM.

Unfortunately we did not have access to data regarding the exact dates when the syndrome diagnoses were confirmed in specific cases.

Reviewer 1: “In the discussion, the part describing "new" genes must be supported by specific clinical evidence that can exclude/include the presence of disease phenotypes. These data are crucial for the clarification of the relationship between high myopia and the different variants identified.” 

R: We have provided a more detailed and clinical evidence-supported discussion regarding "new" genes.

Reviewer 2: “This part should be removed from the statistic paragraph:

"A total of 46 children (24 boys and 22 girls) were included in the study. The mean age of the children at inclusion in the study was 8.9 ± 3.7 years, and the mean refractive error was -9.1 ± 3.1 D. Nine children were lost to follow-up due to unavailability, non-consent to genetic testing, or incomplete genetic analysis."

And included in the results paragraph.” 

R: The suggested corrections have been made: the specified section has been removed from the statistics paragraph and included in the results paragraph.

Reviewer 2: “There is no statistical analysis reported in this study and no statistical tests have been reported but simply the results of the study have been reported.” 

R: We have revised the "Statistics" section to provide more specific details about demographic data and calculation.

Reviewer 2: Have the authors made all data underlying the findings in their manuscript fully available? 

R: All data underlying and supporting the findings can be found in the table included in the manuscript file. Data regarding the genetic alterations and variants was retrieved from the genetic testing reports we accessed at the Clinical Institute of Genomic Medicine’s database in Ljubljana.

Reviewer 2: Improvement of technical aspects of the manuscript 

R: We have improved the technical aspects of the manuscript by reorganizing the Methods and Statistics sections, enhancing the presentation of confirmed cases in the table, and adding specific minor corrections.

Data policy The data that support the findings of this study are shown in the manuscript. The genetic raw data are not publicly available due to privacy and ethical restrictions.

Reviewer:Is the manuscript presented in an intelligible fashion and written in standard English? 

R: Slight improvements in wording were made to make the manuscript more comprehensible. Proofreading of the manuscript was performed by a native English speaker, proficient in proofreading of medical papers.

---

## [Editor Report · Decision Letter 1]

21 Oct 2024

Genetic background of high myopia in children

PONE-D-24-12970R1

Dear Dr. Manca Tekavčič Pompe,

We’re pleased to inform you that your manuscript has been judged scientifically suitable for publication and will be formally accepted for publication once it meets all outstanding technical requirements.

Kind regards,

Giuseppe Novelli

Academic Editor

PLOS ONE
---

## [Editor Report · Acceptance letter]

24 Oct 2024

PONE-D-24-12970R1 

PLOS ONE

Dear Dr. Tekavčič Pompe, 

I'm pleased to inform you that your manuscript has been deemed suitable for publication in PLOS ONE. Congratulations! Your manuscript is now being handed over to our production team.

Kind regards, 

on behalf of

Prof. Giuseppe Novelli 

Academic Editor

PLOS ONE